# Characteristics, laboratories, and prognosis of severe COVID-19 in the Tokyo metropolitan area: A retrospective case series

**Shusuke Mori**[ID]*, **Tomohiko Ai, Yasuhiro Otomo**[ID]

Department of Acute Critical Care and Disaster Medicine, Tokyo Medical and Dental University, Tokyo, Japan

* shunasu@yahoo.co.jp

**Data Availability Statement:** All relevant data are within the manuscript and its Supporting Information file.

## Abstract

The impact of the COVID-19 pandemic has been immense, while the epidemiology and pathophysiology remain unclear. Despite many casualties in many countries, there have been less than 1,000 deaths in Japan as of end of June, 2020. In this study, we analyzed the cases of COVID-19 patients admitted to our institution located in the Tokyo metropolitan area where the survival rate is higher than those in other cities in the world. Medical records of COVID-19 patients that were admitted to a single Japanese tertiary university hospital in the Tokyo metropolitan area between March 10th and June 2nd, 2020 were retrospectively reviewed. The identified COVID-19 cases were subdivided into two groups (severe and mild) depending on the need for mechanical ventilation. Those in the severe group required mechanical ventilation as opposed to those in the mild group. The data were analyzed using nonparametric tests expressed by median [interquartile range (IQR)]. A total of 45 COVID-19 patients were included, consisting of 22 severe cases (Group S) and 23 mild cases (Group M). Male sex (Group S, 95.5% vs. Group M, 56.5%, p<0.01), high body mass index (Group S, 24.89 [22.44–27.15] vs. Group M, 21.43 [19.05–23.75], p<0.01), and hyperlipidemia (Group S, 36.4% vs. Group M, 0%, p<0.01) were more seen in Group S. Five (22.7%) cases in Group S underwent extracorporeal membranous oxygenation (ECMO). On admission, lymphopenia, decreased albumin, and elevated fibrinogen, lactate dehydrogenase, transaminases, creatine kinase, C-reactive protein, and procalcitonin were observed in Group S. The median ICU and hospital stay were 13.5 [10.3–22.3] days and 23.0 [16.3–30.5] days, respectively, in Group S. As of June 28th, 2020, in Group S, 19 (86.4%) patients have survived, of which 17 (77.3%) were discharged, and 2 are still in treatments. Three died of multiple organ failure. All 23 patients in Group M have recovered. Male sex, high body mass index, and hyperlipidemia can be risk factors for severe COVID-19 pneumonia, and its overall short-term survival rate was between 77.3% and 86.4% in this study.

**Funding:** The author(s) received no specific funding for this work.

**Competing interests:** The authors have declared that no competing interests exist.

## Introduction

In late 2019, a new type of corona virus disease (COVID-19) caused by Sever Acute Respiratory Coronavirus 2 (SARS-CoV-2) which originated from Wuhan, China [1], and has spread worldwide, resulting in a pandemic declaration by the World Health Organization (WHO) [2]. According to the latest WHO report (Situation report 141), more than seven million cases were confirmed, and more than 4 million deaths were recorded globally as of June 9th, 2020. In Japan, at least 17,210 PCR-positive cases were confirmed and the death toll rose to 916 according to the WHO report. Despite the lack of preparation and clear guidelines in the medical community in Japan [3], the crude mortality rate in Japan seems to remain lower than those in European countries such as Italy and the United Kingdom (5.3% vs. 14.4%). The mortality rate was 24.5% in hospitalized patients in New York City [4]. Although this might be associated with the difference of ancestral types of SARS-CoV-2 [5] and the possible roles of Bacillus Calmette-Guérin (BCG) vaccination [6, 7], the actual reasons remain totally unknown [3].

Our hospital is located in the metropolitan area of Tokyo, the most populated city in the world with a population of approximately 14.0 million as of January 1st, 2020. The situation in the medical community was chaotic since no guidelines for the treatment of this novel virus existed [3], which was confounded by the hype of news and social media outlets. While many hospitals were forced to shut down to avoid viral transmission among inpatients, our hospital was one of the few tertiary medical centers that treated COVID-19 in Tokyo. In addition, when developing a treatment plan for the patients, we were unaware of the mass misinformation being published and distributed pertaining to risk factors and treatments [8–10].

In this retrospective study, we sought to describe the clinical characteristics, treatments including various medications, extracorporeal membrane oxygenation (ECMO), and short-term outcomes of the 45 patients admitted to our institution from March 10th to June 2nd, 2020. We also discussed treatment costs including extracorporeal membrane ECMO use.

## Methods

### Patients and admission criteria

This study was conducted in accordance with the ethical principles stated in the Declaration of Helsinki and was approved by the institutional review board of Tokyo Medical and Dental University (ethical approval number: M2020-089). Written informed consents were obtained from patients or guardians upon admission. Our tertiary university hospital is located in the center of Tokyo metropolis area where the population is approximately 3.5 million and serviced by 5 tertiary emergency medical centers. The Tokyo metropolitan government designated two hospitals including ours out of five as COVID-19 specialized treatment facilities. As a result, between March 10th and June 2nd, 2020, a total of 45 admitted patients tested positive through polymerase chain reaction (PCR) for COVID-19 and had evidence of pneumonia on computed tomography (CT). Tables 1 and 2 summarize the patient background and initial laboratory findings.

All patients in this study showed positive results for PCR diagnostic tests of nasopharyngeal samples using the method developed by the National Institute of Infectious Diseases in Japan. The diagnosis of pneumonia was confirmed by respiratory medicine specialists based on CT findings.

### Treatments

**Medications.** All medications used in this study are readily available for clinical use in Japan. However, the level of efficacy of these medications on COVID-19 has not been determined [11]. Thus, treatment regimen was not consistent through the study period (Table 3).

**Table 1. Characteristics of the patients with COVID-19 pneumonia.**

| | Total (n = 45) | Group S (n = 22) | Group M (n = 23) | *p* value |
|---|---|---|---|---|
| age (years) ± SD | 64 (48.5–77) | 58 (51.5–76) | 69 (45–83) | NS |
| sex | | | | *p*<0.01 |
| male | 34 (75.6%) | 21 (95.5%) | 13 (56.5%) | |
| female | 11 (24.4%) | 1 (4.5%) | 10 (43.5%) | |
| smoking history | 20 (44.4%) | 8 (36.4%) | 12 (52.3%) | NS |
| body mass index (BMI) | 23.49 (21.09–25.56) | 24.89 (22.44–27.15) | 21.43 (19.05–23.75) | *p*<0.01 |
| comorbidities | | | | |
| cardiovascular all | 21 (46.7%) | 11 (50.0%) | 10 (43.5%) | NS |
| hypertension | 18 (40.0%) | 11 (50.0%) | 7 (30.4%) | NS |
| respiratory all | 11 (24.4%) | 5 (22.7%) | 6 (26.1%) | NS |
| bronchial asthma | 7 (15.6) | 2 (9.1%) | 5 (21.7%) | NS |
| COPD | 1 (2.2%) | 1 (4.5%) | 0 (0.0%) | NS |
| pneumonia/pleuritis | 4 (8.9) | 2 (9.1%) | 2 (8.7%) | NS |
| hyperlipidemia | 8 (17.8%) | 8 (36.4%) | 0 (0.0%) | *p*<0.01 |
| diabetes | 5 (11.1%) | 4 (18.2%) | 1 (4.3%) | NS |
| GERD | 2 (4.4%) | 0 (0.0%) | 2 (8.7%) | NS |
| CKD | 3 (6.7%) | 0 (0.0%) | 3 (13.0%) | NS |
| malignancy | 6 (13.3%) | 4 (18.2%) | 2 (8.7%) | NS |
| symptoms | | | | |
| fever | 40 (88.9%) | 20 (90.9%) | 20 (87.0%) | NS |
| sore throat | 4 (8.9%) | 1 (4.5%) | 3 (13.0%) | NS |
| dyspnea | 16 (35.6%) | 11 (50.0%) | 5 (21.7%) | NS |
| cough | 14 (31.1%) | 8 (36.4%) | 6 (26.1%) | NS |
| taste disturbance | 9 (20%) | 1 (4.5%) | 8 (34.8%) | *p*<0.05 |
| malaise | 12 (26.7%) | 7 (31.8%) | 5 (21.7%) | NS |
| period from onset of symptom to PCR positive (days) | 7 (5–9) | 6.5 (4–8) | 9 (6.5–13) | *p*<0.01 |

Values are presented as median (IQR: interquartile range). Group S consists of severe cases requiring mechanical ventilation and intensive care. Group M consists of moderate cases not requiring mechanical ventilation support managed on the general wards. Cardiovascular diseases include congestive heart failure, unstable angina pectoris, atrial fibrillation, and hypertension. Respiratory diseases include bronchial asthma, chronic obstructive pulmonary disease (COPD), tuberculosis, pleuritis, and pneumonia. Malignancy includes colonic cancer, spinal tumor, brain tumor, prostate cancer, oral cancer, and malignant lymphoma. CKD: Chronic kidney disease; GERD: gastroesophageal reflux disease; PCR: polymerase chain reaction; NS: not significant.

Favipiravir [12], an anti-influenza reagent, was given to all patients orally or via nasogastric tube. The loading dose was 3.6 gm/day, with a maintenance dose was 0.8 gm/day for up to 14 days. Ciclesonide [13], an inhaler bronchodilator, was given to mild to moderate patients. Hydroxychloroquine [14] was given to the severe patients and its loading dose was 800 mg/day followed by a maintenance dose, 400 mg/day for 4 days. Tocilizumab [15], an anti-interleukin 6 receptor monoclonal antibody, was used for severe patients at a one-time dose of 8 mg/kg. Methylprednisolone, used in interstitial pneumonia, was used for severe patients at a dose of 500–1,000 mg for 3 days, and then tapered. The decision or recommendation for each individual drug use was given by the respiratory medicine team depending on the severity of the pneumonia and the CT findings.

**Mechanical ventilation.** Tracheal intubation criteria were (1) low oxygen concentration (90<$SpO_2$), (2) hemodynamically unstable, and (3) tachypnea with respiratory efforts. Patients were placed on mechanical ventilators (Evita® V300, Dräger, Lübeck, Germany) after endotracheal intubation. Positive end expiratory pressure (PEEP) was set to not too excessive (<20

**Table 2. Laboratory findings on admission of the patients with COVID-19 pneumonia comparing severe with moderate cases.**

|  | Group S (n = 22) | Group M (n = 23) | *p* value |
|---|---|---|---|
| CBC |  |  |  |
| white blood cells ($\times 10^3/\mu L$) | 6.85 (5.73–8.45) | 5.5 (4.8–6.6) | *p*<0.05 |
| lymphocytes ($\times/\mu L$) | 715.8 (586.4–1107) | 1189.4 (892.4–1584) | *p*<0.05 |
| platelets ($\times 10^4/\mu L$) | 19.7 (16.3–28.5) | 27.5 (17.3–33.6) | NS |
| coagulation profile |  |  |  |
| PT-INR | 1.2 (1.10–1.37) | 1.08 (1–1.15) | *p*<0.05 |
| APTT (seconds) | 35.6 (31.2–40.5) | 32 (30.8–38.3) | NS |
| fibrinogen (mg/dl) | 558 (505.5–597.5) | 414 (296.5–483.0) | *p*<0.001 |
| FDP (µg/ml) | 8.15 (6.15–16.6) | 7.2 (5.5–8.5) | NS |
| D-dimer (µg/ml) | 1.62 (0.89–9.01) | 1.02 (0.40–2.62) | NS |
| biochemistry |  |  |  |
| albumin (g/dl) | 2.6 (1.88–3.00) | 3.4 (2.8–4.2) | *p*<0.01 |
| BUN (mg/dl) | 17 (10.8–21.0) | 12 (9–17.3) | NS |
| creatinine (mg/dl) | 0.81(0.63–1.18) | 0.81 (0.64–0.87) | NS |
| LDH (U/l) | 439.5 (322.8–534.5) | 234 (190–302) | *p*<0.001 |
| AST (U/l) | 67 (41.5–102.5) | 24 (19–47) | *p*<0.001 |
| ALT (U/l) | 53 (27–76.3) | 19 (10–27) | *p*<0.01 |
| CK (U/l) | 89 (40–289) | 48 (35–79) | *p*<0.05 |
| serum |  |  |  |
| CRP (mg/dl) | 11.5 (6.62–20.0) | 2.13 (0.08–4.00) | *p*<0.001 |
| procalcitonin (ng/dl) | 0.24 (0.09–1.19) | 0.06 (0.03–0.08) | *p*<0.001 |

Values are presented as median (IQR: interquartile range). PT-INR: prothrombin time-international normalized ratio; APTT: activated partial thromboplastin time; FDP: fibrin degradation products; BUN: blood urea nitrogen; LDH: lactate dehydrogenase; AST: aspartate aminotransferase; ALT: alanine aminotransferase; CK: creatine kinase; CRP: C-reactive protein; NS: not significant.

$cmH_2O$) to avoid barotrauma. Patients were placed on prone position, if necessary [16]. The extubation criteria were (1) awake and oriented, (2) capable of expectoration, (3)hemodynamically stable, (4) showing normal blood gas, (5) respiratory rate ≤35 per minute, (6) tidal volume >5 ml/kg, (7) $FiO_2$ ≤0.4, PEEP ≤8$cmH_2O$, $SpO_2$ >90% or P/F >150, and (8) RSBI (RR/ VT) <100/min/L.

**Venovenous extracorporeal membrane oxygenation (VV-ECMO).** VV-ECMO (SP-200 NEO, Terumo Corp., Tokyo) was used in severe cases of respiratory distress where the patient could not maintain their respiratory function even while fully supported by mechanical ventilation. The indication criteria in COVID-19 cases were as follows; (1) $PaO_2/FaO_2$ ratio (P/F) <80 mmHg for >6 hours, (2) P/F <50 mmHg for >3 hours, (3) pH <7.25 with PaCO2 ≥60 mmHg for >6 hours after making every effort to try prone position, muscle relaxant, high positive end-expiratory pressure (PEEP) strategy, nitric oxide inhalation, and recruitment maneuver, and (4) P/F ≥150 mmHg and pH <7.25 with $PaCO_2$ ≥60 mmHg for 6 hours. The exclusion criteria were (1) not consented, (2) hemorrhagic complications, (3) severe disturbance of central nervous system, (4) end-of-life stage, and (5) ≥75-year-old. Our withdrawal criteria were similar to the interim Extracorporeal Life Support Organization (ELSO) guideline [17].

**Anticoagulation.** COVID-19 has been reported to cause a hypercoagulation state which sometimes manifests clinically as microvasculitis and biochemically high FDP or D-dimer

**Table 3. Treatments, complications, clinical course, and medical costs for the patients with PCR-positive for SARS-CoV-2 managed on ventilator in intensive care units.**

| age | sex | medications | complications | period on ventilator (days) | ECMO | period on ECMO (days) | CRRT (CHDF) | prone position | recovery | ICU stay (days) | total admission days | total costs (USD) |
|---|---|---|---|---|---|---|---|---|---|---|---|---|
| 55 | M | FV, CS, HCQ, HepCa | liver dysfunction | 9 | | | | | + | 18 | 32 | 28,429 |
| 64 | M | FV, HCQ, mPSL, Hep | | 7 | | | | | + | 8 | 21 | 16,611 |
| 57 | M | FV, NM, TZB, HCQ, rTM, Hep | pneumothorax, hemothorax, shingles | 52 | + | 10 | + | + | + | 61 | 61 | 127,772 |
| 50 | M | FV, HCQ, HepCa | bilateral pneumothorax | 11 | | | | | + | 16 | 29 | 29,137 |
| 77 | M | FV, mPSL, HCQ, TZB, rTM, Hep | | 11 | | | | | | 11 | 11 | 25,995 |
| 66 | M | FV, TZB, HCQ, mPSL, Hep | pneumothorax, hemothorax, hemorrhage in the chest wall, shingles | 65 | + | 62 | + | + | + | 65 | 65 | 182,586 |
| 30 | F | FV, mPSL, TZB, IVIg, NM, Hep | | 16 | + | 9 | | + | + | 20 | 20 | 69,123 |
| 76 | M | FV, TZB, NM, Hep | | 17 | | | | + | + | 23 | 23 | 41,487 |
| 53 | M | FV, HP, HCQ, NM, Hep | | 20 | + | 10 | | | + | 23 | 31 | 66,712 |
| 28 | M | FV, HCQ, NM, AT, Hep | rhabdomyolysis, pneumomediastinum, non-occlusive mesenteric ischemia | 35 | | | + | + | | 35 | 35 | 91,622 |
| 54 | M | FV, CS, HCQ, NM, TZB, rTM, Hep | | 47 | + | 22 | | + | + | 55 | 55 | 124,443 |
| 84 | M | FV, NM, HCQ, TZB, rTM, HC, Hep | | 17 | | | | + | | 17 | 17 | 23,475 |
| 46 | M | FV, mPSL, TZB, Hep | | 9 | | | | | + | 5 | 5 | 13,667 |
| 43 | M | FV, CS, mPSL, TZB, Hep | pulmonary embolism | 8 | | | | + | + | 10 | 29 | 30,501 |
| 76 | M | FV, TZB, NM, Hep | | 11 | | | | + | + | 14 | 27 | 37,852 |
| 68 | M | FV, CS, XR, TZB, mPSL, rTM, Hep | pulmonary embolism, deep venous thrombosis | 5 | | | | + | + | 7 | 7 | 24,475 |
| 77 | M | FV, TZB, rTM, Hep | hemorrhage in the chest wall | 5 | | | | | + | 13 | 29 | 45,915 |
| 84 | M | FV, TZB, INH, mPSL, Hep | ST elevation myocardial infarction, upper gastrointestinal bleeding | 13 | | | | + | + | 13 | 13 | N/A |
| 52 | M | FV, HCQ | | 5 | | | | | + | 7 | 16 | 22,637 |
| 59 | M | FV, mPSL, HCQ, Hep | | 5 | | | | + | + | 6 | 18 | 22,253 |
| 73 | M | FV, Hep | | 11 | | | | | + | 11 | 11 | 33,099 |

(*Continued*)

**Table 3.** (Continued)

| age | sex | medications | complications | period on ventilator (days) | ECMO | period on ECMO (days) | CRRT (CHDF) | prone position | recovery | ICU stay (days) | total admission days | total costs (USD) |
|---|---|---|---|---|---|---|---|---|---|---|---|---|
| 52 | M | FV, CS, HCQ, AT, HepCa | | 9 | | | | | + | 12 | 23 | 35,040 |

ECMO: Extracorporeal membrane oxygenation; CHDF: Continuous hemodiafiltration; CRRT: Continuous renal replacement therapy; ICU: intensive care unit; USD: United States dollar; AT: antithrombin III; CS: ciclesonide; FV: favipiravir; HC: hydrocortisone; HCQ: hydroxychloroquine; Hep: intravenous heparin sodium; HepCa: subcutaneous heparin calcium; INH: isoniazid; IVIG: intravenous immunoglobulin; mPSL: methylprednisolone; NM: nafamostat mesylate; rTM: human recombinant thrombomodulin; TZB: tocilizumab; XR: rivaroxaban; N/A: not available.

[18]. Therefore, anticoagulants were given unless otherwise contraindicated. Intravenous heparin as well as nafamostat mesylate, recombinant human thrombomodulin, and antithrombin III were given referring to activated partial thromboplastin time (APTT) and activated clotting time (ACT). When the patients were on VV-ECMO and/or CRRT, anticoagulation including heparin was mandatory.

**Prone position.** The criteria for placing a patient in the prone position are the following. The prone position is considered when (1) the patient's P/F ratio goes down to below 200; (2) the CT shows severe inflammation and/or consolidation mainly on the dorsal side of the lung. The prone position is considered contraindicated when (1) the patient is unable to tolerate prone position due to hemodynamic instability; (2) the patient has any complications such as internal or external hemorrhages and severe skin damages.

## Statistics

Continuous variables were expressed as median and interquartile range (IQR). Comparison of the data between two groups was performed by Mann–Whitney U tests, Fisher's exact test, and chi-square test using SPSS® Statistics Version 25 (IBM®, Armonk, New York). The value of $p < 0.05$ was considered as statistically significant.

## Results

### Baseline patient characteristics

A total of 45 patients were admitted: 22 patients with severe pneumonia admitted to the intensive care unit and treated with mechanical ventilation (Group S) and 23 patients with moderate pneumonia admitted to regular wards without mechanical ventilation (Group M). Table 1 summarizes the baseline characteristics. The median age was not significantly different between the two groups. Male sex was predominant in Group S (95.5%) compared to Group M (56.5%). (p<0.01) All other cases were community-acquired infections. The body mass indices were significantly higher in Group S than Group M (24.89 vs. 21.43, p<0.01). There were no differences in smoking habits, pre-existing cardiovascular diseases, respiratory disease, and diabetes mellitus. In contrast, the prevalence of hypercholesterolemia was significantly higher in Group S than Group M (36.4% vs. 0%, $p < 0.01$). Prodromal symptoms were quite similar between the two groups except the rate of taste disturbance (4.5% vs. 34.8%, p<0.05). The average duration from the onset of the symptoms to confirmation of positive for COVID-19 PCR tests was approximately 2.5 days shorter in Group S than Group M (p<0.01), indicating Group S patients possibly became ill quicker than Group M patients.

Table 2 summarizes the result of laboratory examination on admission. Whereas white blood cell counts were higher in Group S (6.85 vs. $5.5 \times 10^3$/μL, p<0.05), lymphocytes were

lower in Group S than Group M (715.8 vs. 1189.4×10/μL, p<0.05). Blood biochemistry showed higher prothrombin time-international normalized ratio (PT-INR) (1.2 vs 1.08, p<0.05), higher fibrinogen (558 vs. 414 mg/dl, p<0.001), lower albumin (2.6 vs. 3.4 g/dl, p<0.01), higher lactate dehydrogenase (LDH) (439.5 vs. 234 U/l, p<0.001), higher aspartate aminotransferase (AST) (67 vs. 24 U/l, p<0.001), higher alanine aminotransferase (ALT) (53 vs. 19 U/l), higher creatine kinase (CK) (89 vs. 48 U/l, p<0.05), higher c-reactive protein (CRP) (11.5 vs. 2.13 mg/dl, p<0.001), and higher procalcitonin (0.24 vs. 0.06, p<0.001) in Group S. Group S patients showed marked lymphopenia in the early phase of the treatment.

## Treatments

Table 3 summarizes the treatments for Group S patients. Two patients were already intubated when transferred from other hospitals (5 and 14 days before admission). All Group S patients were given combination of medications, hydroxychloroquine (54.5%), tocilizumab (54.5%), ciclesonide (22.7%), methylprednisolone (40.9%), and anticoagulants. The median period for mechanical ventilation treatment was 11.0 days [8.25–17.0]. Thirteen patients (59.1%) were treated in prone position. Five patients (22.7%) received ECMO treatment. Three patients (13.6%) received continuous renal replacement therapy (CRRT), equivalent to continuous hemodiafiltration (CHDF). The median period for ECMO treatment was 10.0 days [10.0–22.0]. The median period of ICU stays and the median length of hospital stays were 13.5 days [10.25–22.25] and 23 days [16.25–30.5], respectively. As for Group M patients, most of them received ciclesonide and/or favipiravir, and none required mechanical ventilation (S1 Table). The median hospital cost for the patients in Group S was approximately $33,099 [24,475–66,712]. During their respective clinical course, three patients in Group S were found to have pneumothorax most likely associated with positive airway pressure. Two of them had hemothorax which may have been related to anticoagulant therapy. Two patients had pulmonary embolism and deep vein thrombosis. One male patient developed ST-elevation myocardial infarction (STEMI), and he subsequently had a successful percutaneous coronary artery intervention.

## Short-term outcomes

Table 4 summarizes the overall outcomes.

Seventeen (77.3%) patients in Group S survived and were discharged. All Group M patients have recovered and discharged. Currently, two patients are still receiving mechanical ventilation, and one is on ECMO (62 days). Three patients in Group S that died are: a 77-year-old male died of multiple organ failure (MOF) at day 11; a 28-year-old male died of MOF associated with rhabdomyolysis-induced acute kidney failure and necrotic enteritis at Day 35; and an 84-year-old male died of acute kidney failure and respiratory failure at Day 17.

Table 4. Modalities of treatment and overall short-term outcome of COVID-19 pneumonia patients.

| | total (n = 45) | Group S (n = 22) | Group M (n = 23) |
|---|---|---|---|
| mechanical ventilation | 22 (48.9%) | 22 (100.0%) | 0 (0%) |
| ECMO | 5 (22.7%) | 5 (22.7%) | 0 (0%) |
| CRRT | 3 (13.6%) | 3 (13.6%) | 0 (0%) |
| prone position | 13 (28.9%) | 13 (59.1%) | 0 (0%) |
| survived and discharged | 40 (88.9%) | 17 (77.3%) | 23 (100%) |
| deceased | 3 (6.7%) | 3 (13.6%) | 0 (0%) |
| in treatment | 2 (4.4%) | 2 (9.1%) | 0 (0%) |

ECMO: Extracorporeal membrane oxygenation; CRRT: continuous renal replacement therapy.

## Discussion

This is one of the few reports describing COVID-19 pneumonia from Tokyo, Japan. Tokyo is a very populated metropolitan area in the world, and extremely high casualties were expected as was reported in many other countries [19]. However, the total death toll by June 2020 is less than 1,000 in Japan, which is very unexpected. The epidemiology and pathophysiology of COVID-19 remains totally unelucidated at this moment. Although several reports claimed the sequencing of SARS-CoV-2 RNA has been determined [5, 20], how and where this RNA virus metamorphoses have not yet been fully understood. Clinical phenotypes also widely vary depending upon the regions as shown in hundreds of case reports: from asymptomatic to severe acute respiratory failure, multiple organ failure, and systemic thrombosis [1, 21]. There-fore, this report can provide beneficial information to further understanding of this pandemic.

### Treatments and clinical outcomes

In the initial report from Wuhan, China, older age, higher D-dimer, and higher Sequential Organ Failure Assessment (SOFA) scores on admission were associated with poor prognosis [1]. According to the study in New York, older age, male gender, and pre-existing hyperten-sion and diabetes were highly prevalent in the hospitalized patient cohort [4]. In our patients, male gender, high body mass index (BMI) and hyperlipidemia were more prevalent in the severe pneumonia group compared to the moderate pneumonia group. However, our patient cohort is relatively small to make clear conclusions. Furthermore, it is difficult to predict who will develop severe respiratory failure that will require ECMO. Extent of immune reactions, i.e., gene expression levels in immune cells, may be one of the important determining factors of severity as shown in a single-cell analysis of bronchoalveolar immune cells [22]. Indeed, in our patients, consistent with the recent reports [23], lymphopenia was more observed in Group S than Group M. In addition, the comorbid lipid disorders were more prevalent in Group S patients, which might be associated with abnormal inflammatory processes caused by deteriorated lipid metabolisms [24, 25].

Although there was no clear evidence to treat SARS-CoV-2 infection hydroxychloroquine / chloroquine [14], ivermectin [26], lopinavir / ritonavir [27], and anti-interleukin agents were reported to be potentially effective [11]. Since thrombotic disorders were also reported as one of the critical manifestations [28, 29], we used various combinations of drugs and anticoagu-lant therapies in Group S patients (Table 3). Group M patients received favipiravir and cicleso-nide, and all of them successfully recovered; however, we cannot make any conclusions regarding the drugs' efficacy from this study due to lack of randomized, placebo control design.

Overall, the survival rates in our patients were higher than the previous case series in other countries including the U.S. and China [1, 4, 30]. The survival rate of the patients on ECMO was not as dire as it was claimed in several reports [31, 32]. However, currently we do not know what kind of factors made these differences. Further analyses of large data are warranted.

**Complications.** During the treatment period for the severe COVID-19 pneumonia cases, we experienced some remarkable complications. First, we had three pneumothorax cases in addition to one case of pneumomediastinum and subcutaneous emphysema among only 22 severe COVID-19 cases. There have so far been some case reports, all of which describe they are rare complications [33–36]; however, from this study it is unlikely that we can say they are rare. All cases demonstrating pneumothorax were treated by thoracostomy with trocar inser-tion and continuous low negative pressure. Two pneumothorax cases were managed by lung

rest strategy using ECMO. In refractory cases of pneumothorax, surgical intervention with thoracoscopy may be considered if conservative treatment fails [36].

Another remarkable complication was hypercoagulation disorders. We had two pulmonary embolisms and deep venous thrombosis, and one ST elevation myocardial infarction, which were suspected to be tightly related to COVID-19. In addition, FDP and D-dimer significantly rose up in most severe COVID pneumonia cases, indicating hypercoagulable state, which required us to keep the blood anticoagulated using antithrombin III, human recombinant thrombomodulin, nafamostat mesylate, and/or heparin to avoid coagulopathic complications. Furthermore, we also had hemorrhagic complications probably due to such anticoagulation therapy.

**Impact on the financial status.** An initial case series from China did not show significant role of ECMO in the treatment of COVID-19 pneumonia [37]. Although the exact costs were not shown in the paper, the investigators are skeptical about the cost effectiveness of ECMO. A case series in Norway reported that the mean estimated cost for ECMO procedures was $73,122, ranging from $59,871 to 405,497, even in 2010 [38]. In the U.S., the costs for ECMO procedure used to be $5,000-$10,000 per day [39], and the total in-hospital costs could range from $42,554 to $537,554 in 2013 depending upon the underlying diseases [40]. Current costs and patient charges for ECMO procedures in the U.S. might be far higher than these. These high costs raised concerns in the medical societies in many countries [32, 41].

The medical expense in Japan is relatively cheap compared to the Western countries according to the OECD Data [42]. Furthermore, everyone residing in Japan is covered by social insurance and therefore has a right to take the best available therapy regardless of the type of health insurance. These facts may contribute to the better survival rate. The total cost for ECMO equipment is approximately $3,527 (388,000 Japanese yen; $0.901/yen). The cost for introductory procedure is $1,009, and maintenance and trouble-shooting cost only $300/day. The cost for ICU admission is $1,291/day for the 1st week, then reduced to $1,148/day for the 2nd week. After 2 weeks, ICU admission fee cannot be charged. The cost for mechanical ventilation management is $74/day. The median cost for Group S was $31,800, which will heavily impact the medical costs if the number of admitted COVID-19 patients increases.

## Limitations

This study has several significant limitations: (1) this is a single center study with a small number of patients. However, our institution served a very populated area and the patients' clinical profiles might be a representative sample of Tokyo population. In addition, we have had a very limited number of COVID-19 patients admitted nationwide in Japan as opposed to other countries, and in terms of severe cases, the number has been all the more limited; (2) the diagnoses of SARS-CoV-2 positive merely relied upon limited PCR methods that can cross react with other pathogens [43]; (3) the effects of all medications cannot be validated in this study due to the study design, i.e., non-placebo controlled, randomized study; (4) we did not describe in-depth coagulation disorders in this study. Those are currently under analyses, and will be reported separately; (4) remdesivir was not used in this study since it has not yet been approved for clinical use in Japan; and (5) we do not know the long-term prognosis due to lack of adequate follow up periods.

## Conclusion

We treated a total of 45 moderate to severe COVID-19 pneumonia patients in one of the most advanced facilities in a metropolitan area in Tokyo, making the best of the most updated

treatment and management available worldwide at that moment. Overall short-term survival rate among severe COVID-19 pneumonia cases ranged from 77.3% to 86.3% in this study.

## Supporting information

**S1 Table. Characteristics, treatments, and prognosis in Group M.** BMI: body mass index; GERD: gastroesophageal reflux disease; CKD: chronic kidney disease; CHF: congestive heart failure; DM: diabetes mellitus; DVT: deep venous thrombosis; CS: ciclesonide; FV: favipiravir; XR: Xarelto (rivaroxaban); Hep: heparin; mPSL: methylprednisolone; N/A: not available. (XLSX)

## Acknowledgments

The authors are grateful to the patients who consented to be part of this study, and all physicians, paramedics, and staffs who have been conducting self-sacrificing duties against the pandemic despite very discouraging circumstances. The authors also thank to Dr. Corina Rosales and Dr. Jean Dominique Morancy for their critical reading of the manuscript.

## Author Contributions

**Conceptualization:** Shusuke Mori.

**Data curation:** Shusuke Mori.

**Formal analysis:** Shusuke Mori.

**Investigation:** Shusuke Mori.

**Methodology:** Shusuke Mori, Tomohiko Ai.

**Project administration:** Shusuke Mori, Tomohiko Ai.

**Supervision:** Yasuhiro Otomo.

**Validation:** Tomohiko Ai.

**Writing – original draft:** Shusuke Mori, Tomohiko Ai.

**Writing – review & editing:** Tomohiko Ai.

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
