## [Decision Letter · Decision Letter 0]

25 Aug 2020

PONE-D-20-19930

Characteristics, laboratories, and prognoses of severe COVID-19 in the Tokyo metropolitan area: a retrospective case series

PLOS ONE

Dear Dr. Mori,

Thank you for submitting your manuscript to PLOS ONE. After careful consideration, we feel that it has merit but does not fully meet PLOS ONE’s publication criteria as it currently stands. Therefore, we invite you to submit a revised version of the manuscript that addresses the points raised during the review process.

Both of the reviewers think your paper important, so please response the problems pointed by the review.

We look forward to receiving your revised manuscript.

Kind regards,

Yoshiaki Taniyama, MD, PhD

Academic Editor

PLOS ONE

Journal Requirements:

Reviewers' comments:

Reviewer's Responses to Questions

**Comments to the Author**

1. Is the manuscript technically sound, and do the data support the conclusions?

Reviewer #1: Yes

Reviewer #2: Yes

2. Has the statistical analysis been performed appropriately and rigorously? 

Reviewer #1: Yes

Reviewer #2: Yes

3. Have the authors made all data underlying the findings in their manuscript fully available?

Reviewer #1: Yes

Reviewer #2: Yes

4. Is the manuscript presented in an intelligible fashion and written in standard English?

Reviewer #1: Yes

Reviewer #2: Yes

5. Review Comments to the Author

Reviewer #1: This is a timely and well done review of the Tokyo experience with COVID-19. It provides some answers but mostly raises some helpful questions regarding the pathophysiology of COVID-19 PNA. Particularly informative was the discussion section where the authors compare the Tokyo experience to that in NYC and Italy. It seems that elevated BMI and hyperlipidemia are pertinent risk factors for COVID-19 PNA.

I would suggest editing of this paper by an individual who speaks English as their first and primary language. There were a number of spelling errors including "Severe" in the very first sentence.

Also, can you explicitly state the criteria for placing a patient in the prone position?

Seeing at the patients were all anticoagulated, were any bleeding complications encountered? eg: epistaxis, hemoptysis. Were INR, PT and PTT monitored? What was the range of these values in those patients who recovered?

Procalcitonin is misspelled- page 14

Ventilator is misspelled table 4.

Overall nice paper. Will add to the medical literature. With these few revisions it should be ready for publication.

Reviewer #2: This manuscript reports characteristics, laboratories, treatment associated cost in ICU of hospitalized COVID-19 patients in the Tokyo metropolitan area. It is a retrospective study with 45 patients, consisting 22 severe cases and 23 mild cases. Descriptive statistics and comparisons between 22 severe cases and 23 mild cases were performed with appropriated statistical methods. Even though the sample size is small, these data may provide valuable information about the observed differences in characteristics, laboratories between severe cases and mild cases. And also reveals the treatment cost in Japan. Below are my questions.

Page 4, 1st paragraph of Patients and admission criteria, should “April 10” be “March 10” as the start time point?

Page 15, Table 4 shows 17( 77.3%) survived in S group. Should 77.3% be discharged rate because the 1st line of page 16 indicates a 86.3% survival rate? Please make those numbers, labels and descriptions be consistent.

6. PLOS authors have the option to publish the peer review history of their article (what does this mean?). If published, this will include your full peer review and any attached files.

Reviewer #1: No

Reviewer #2: No

---

## [Author Response · Author response to Decision Letter 0]

3 Sep 2020

Response to reviewers

We greatly appreciate your kind review of our manuscript entitled “Characteristics, laboratories, and prognosis of severe COVID-19 in the Tokyo metropolitan area: A retrospective case series” and precious comments’ which are of much help to improve our manuscript. We have carefully read the reviewers’ and the academic editor’s comments and made deliberate revisions. The following are our responses (in black) to their comments (in blue) with the revised sentences (in red). 

Reviewer #1: This is a timely and well done review of the Tokyo experience with COVID-19. It provides some answers but mostly raises some helpful questions regarding the pathophysiology of COVID-19 PNA. Particularly informative was the discussion section where the authors compare the Tokyo experience to that in NYC and Italy. It seems that elevated BMI and hyperlipidemia are pertinent risk factors for COVID-19 PNA.

Response: We are very grateful for your kind review. We revised our manuscript according to your suggestions and made a response to each comment. 

Comment: I would suggest editing of this paper by an individual who speaks English as their first and primary language. There were a number of spelling errors including "Severe" in the very first sentence.

Response: Thank you for your kind suggestion. We have had our manuscript proofread by a doctor, Jean Dominique Morancy, MD, MPH, MBA, who speaks English as a primary language. All the revisions and corrections of grammatical and spelling errors are reflected in the revised manuscript with track changes. Moreover, we expressed our gratitude to the doctor adding the doctor’s name to the acknowledgements section.

Comment: Also, can you explicitly state the criteria for placing a patient in the prone position?

Response: Thank you for your comment. We made a revision regarding the criteria for placing a patient in the prone position as follows.

Page 12-13 (with track changes): added;

Prone Position

The criteria for placing a patient in the prone position are the following. The prone position is considered when (1) the patient’s P/F ratio goes down to below 200; (2) the CT shows severe inflammation and/or consolidation mainly on the dorsal side of the lung. The prone position is considered contraindicated when (1) the patient is unable to tolerate prone position due to hemodynamic instability; (2) the patient has any complications such as internal or external hemorrhages and severe skin damages.

Comment: Seeing at the patients were all anticoagulated, were any bleeding complications encountered? eg: epistaxis, hemoptysis. Were INR, PT and PTT monitored? What was the range of these values in those patients who recovered?

Response: We appreciate your comment. Yes, there are some bleeding complications encountered during the period of this study. The types of complications are hemothorax and chest wall hemorrhage as shown in Table 3. We also described this matter in the last paragraph of ‘Complications’ in the discussion section in page 19-20 (with track changes). Therefore, we did not add any more information to the original text.

In patients with severe COVID-19 and on anticoagulation therapy, especially on heparin, INR, PT and PTT were monitored to avoid hemorrhagic complications due to excessive anticoagulation. However, there was no information regarding the range of these values in those who recovered. 

Comment: Procalcitonin is misspelled- page 14. Ventilator is misspelled table 4.

Response: We greatly appreciate your pointing out our mistakes. We had the entire manuscript proofread by Dr. Jean Dominique Morancy, and made corrections of grammatical and spelling errors.

Comment: Overall nice paper. Will add to the medical literature. With these few revisions it should be ready for publication. 

Response: Thank you for your kind and encouraging comments. We greatly appreciate them. We have made revisions as much in accordance with the comments as we could. We hope our study will be of much help in the research and clinical fields of COVID-19.

Reviewer #2: This manuscript reports characteristics, laboratories, treatment associated cost in ICU of hospitalized COVID-19 patients in the Tokyo metropolitan area. It is a retrospective study with 45 patients, consisting 22 severe cases and 23 mild cases. Descriptive statistics and comparisons between 22 severe cases and 23 mild cases were performed with appropriated statistical methods. Even though the sample size is small, these data may provide valuable information about the observed differences in characteristics, laboratories between severe cases and mild cases. And also reveals the treatment cost in Japan. Below are my questions.

Response: We greatly appreciate your kind and very valuable comments. We made revisions in accordance with them and hope that our responses meet the suggestions you made.

Comment: Page 4, 1st paragraph of Patients and admission criteria, should “April 10” be “March 10” as the start time point?

Response: Yes, it should be. Thank you for pointing out the important point. We revised this as follows.

Page 4 (with track changes);

Previous: between April 10 and June 2, 2020, a total of 45…

Revised: between March 10th and June 2nd, 2020, a total of 45… 

Comment: Page 15, Table 4 shows 17( 77.3%) survived in S group. Should 77.3% be discharged rate because the 1st line of page 16 indicates a 86.3% survival rate? Please make those numbers, labels and descriptions be consistent.

Response: We greatly appreciate your comment. As you suggested, the numbers were not consistent. We made revisions as follows;

Page 15-16 (with track changes), Table 4;

“survived” was changed to “survived and discharged”

Page 17 (with track changes);

Previous: Nineteen Group S patients (86.3%) survived and 17 were discharged.

Revised: Seventeen (77.3%) patients in Group S survived and were discharged. 

Page 2; Abstract-Results

Previous: 19 Group S patients (86.4%) survived, 17 (77.3%) were discharged, and 2 are still in treatments. In Group S, 3 died of multiple organ failure.

Revised: in Group S, 19 (86.4%) patients have survived, of which 17 (77.3%) were discharged, and 2 are still in treatments. Three died of multiple organ failure.

Response to the academic editor, regarding journal requirements

Response: Thank you for your suggestion. The phrase “data not shown” was inappropriate. We added a Supporting Information file to the submission system with a caption and a legend in the manuscript.

---

## [Decision Letter · Decision Letter 1]

11 Sep 2020

Characteristics, laboratories, and prognosis of severe COVID-19 in the Tokyo metropolitan area: A retrospective case series

PONE-D-20-19930R1

Dear Dr. Mori,

We’re pleased to inform you that your manuscript has been judged scientifically suitable for publication and will be formally accepted for publication once it meets all outstanding technical requirements.

Kind regards,

Yoshiaki Taniyama, MD, PhD

Academic Editor

PLOS ONE

Reviewers' comments:

Reviewer's Responses to Questions

**Comments to the Author**

1. If the authors have adequately addressed your comments raised in a previous round of review and you feel that this manuscript is now acceptable for publication, you may indicate that here to bypass the “Comments to the Author” section, enter your conflict of interest statement in the “Confidential to Editor” section, and submit your "Accept" recommendation.

Reviewer #2: All comments have been addressed

2. Is the manuscript technically sound, and do the data support the conclusions?

Reviewer #2: (No Response)

3. Has the statistical analysis been performed appropriately and rigorously? 

Reviewer #2: (No Response)

4. Have the authors made all data underlying the findings in their manuscript fully available?

Reviewer #2: (No Response)

5. Is the manuscript presented in an intelligible fashion and written in standard English?

Reviewer #2: (No Response)

6. Review Comments to the Author

Reviewer #2: (No Response)

7. PLOS authors have the option to publish the peer review history of their article (what does this mean?). If published, this will include your full peer review and any attached files.

Reviewer #2: No

---

## [Editor Report · Acceptance letter]

16 Sep 2020

PONE-D-20-19930R1 

Characteristics, laboratories, and prognosis of severe COVID-19 in the Tokyo metropolitan area: A retrospective case series 

Dear Dr. Mori:

I'm pleased to inform you that your manuscript has been deemed suitable for publication in PLOS ONE. Congratulations! Your manuscript is now with our production department. 

Kind regards, 

on behalf of

Dr Yoshiaki Taniyama 

Academic Editor

PLOS ONE